# Modelling of Corrosion-Induced Concrete Cover Cracking Due to Chloride Attacking

**DOI:** 10.3390/ma14061440

**Published:** 2021-03-16

**Authors:** Pei-Yuan Lun, Xiao-Gang Zhang, Ce Jiang, Yi-Fei Ma, Lei Fu

**Affiliations:** 1Guangdong Provincial Key Laboratory of Durability for Marine Civil Engineering, College of Civil and Transportation Engineering, Shenzhen University, Shenzhen 518060, China; lpysci@szu.edu.cn (P.-Y.L.); 1800331001@szu.edu.cn (C.J.); 2School of Civil Engineering, Central South University, Changsha 410000, China; 3The Construction Quality and Safety Supervision Center of Ningxia Hui Autonomous Region, Yinchuan 750001, China; yifeima@126.com; 4Kaifeng Credit Information Center, Kaifeng 475000, China; fulei19862008@126.com

**Keywords:** concrete cover cracking, modified corrosion current density, semi-elliptical defect, stress intensity factor, probabilistic analysis

## Abstract

The premature failure of reinforced concrete (RC) structures is significantly affected by chloride-induced corrosion of reinforcing steel. Although researchers have achieved many outstanding results in the structural capacity of RC structures in the past few decades, the topic of service life has gradually attracted researchers’ attention. In this work, based on the stress intensity, two models are developed to predict the threshold expansive pressure, corrosion rate and cover cracking time of the corrosion-induced cracking process for RC structures. Specifically, in the proposed models, both the influence of initial defects and modified corrosion current density are taken into account. The results given by these models are in a good agreement with practical experience and laboratory studies, and the influence of each parameter on cover cracking is analyzed. In addition, considering the uncertainty existing in the deterioration process of RC structures, a methodology based on the third-moment method in regard to the stochastic process is proposed, which is able to evaluate the cracking risk of RC structures quantitatively and predict their service life. This method provides a good means to solve relevant problems and can prolong the service life of concrete infrastructures subjected to corrosion by applying timely inspection and repairs.

## 1. Introduction

The chloride-induced corrosion of reinforcement bars is considered as one of the dominant causes of the deterioration of reinforced concrete (RC) structures [1,2,3]. In view of its close relation to the service performance and residual life of in-service RC structure [4], it has drawn intensified study from a number of researchers. The corrosion of reinforcing bars caused by chloride is mainly the presence of free chloride ions.

In detail, chloride ions are normally widespread in structures undergoing de-icing programs such as infrastructure buildings, cross-sea bridges and industrial structures, and all structures undergoing de-icing programs, which firstly results in the corrosion of reinforcement and further leads to the appearance and accumulation of corrosion products between steel and concrete. Meanwhile, the rusty strains between steel and concrete are produced.

Next, longitudinal cracks appear from the reinforcement to the concrete surface, which may finally lead to a general failure of the RC structure. Therefore, by taking into account the above assertions, studies on RC structures incorporating concrete cover cracking in chloride-laden environments are endowed with great engineering significance and such efforts are capable of prolonging the service life and reducing the maintenance cost of RC structures [5].

As a fundamental problem with respect to the chloride-induced corrosion of reinforcing bars, concrete cover cracking time is regarded as one of the most essential characteristics in evaluating the service life of RC structures. Specifically, the threshold expansion pressure and corrosion rate of concrete reaches the maximum at the moment of cover cracking, which has been widely studied by researchers [1,6,7,8,9]. In the past thirty years, much effort focused on experimental studies has been directed towards the exploration of cover cracking time induced by the corrosion of reinforcing bars [10,11,12,13,14], along with a variety of corrosion models that were developed and implemented to predict concrete cover cracking time [15,16,17,18,19]. Precisely, among the above models, thick or thin-walled cylinder and elasticity theory are usually utilized for modeling the concrete with embedded steel bars and evaluating the cover cracking time, respectively [6,20,21]. It is worth noting that in those approaches, concrete is succinctly assumed to be homogeneous and isotropic without initial defects. In fact, concrete is a heterogeneous material, which mainly consists of cement, aggregates and chemical admixture. It inevitably contains assorted initial defects such as random cracks. Noticing this fact, Zhang et al. [22] developed a dynamic model based on the fracture mechanics approach by taking into account the initial defects. This model is capable of predicting the initiation time of initial defects, cover cracking time, threshold expansion pressure, and critical corrosion rate of reinforcing bar, which provided a more reasonable prediction associated with the serviceability of RC structures. However, in this model, two essential factors were ignored: the shape of the initial defects and the corrosion current density with time. These two factors are verified to play significant roles on the stresses induced by corrosion products residing in concrete, and subsequently affect the cover cracking time [23,24,25,26,27,28,29]. Therefore, in order to obtain a certain model with high accuracy with regard to the cover cracking problem, the coupling effects of the shape of initial defects and modified corrosion current density should be taken into account. Furthermore, although the aforementioned models [15,16,17,18,19,30,31] are capable of addressing and modelling several problems with regard to concrete cover cracking time, it is worth stressing that the approaches adopted were generally developed within a deterministic framework and in deficiency of delineating the significant uncertainties during the deterioration process of RC structures, which means the applicability is inevitably limited. Hence, it is appropriate to adopt a probabilistic approach for the characterization of the cracking process and predicting the relevant remaining service life of RC structures.

In the current paper, the main objective is to propose a corrosion-induced cracking model considering both the natural environment corrosion and acceleration to power corrosion. In the proposed model, the initial defects in three-dimensional and the modified corrosion current density formula of reinforcement bars are taken into account [32,33]. Formulas for predicting the cover cracking time, threshold expansion pressure and corrosion rate are also developed, respectively. In order to validate its accuracy and validity, comparisons of the predictions are conducted with experimental results obtained in the relevant literature [6,34,35]. In addition, the influences of relevant parameters on cover cracking time, the threshold expansive pressure and corrosion rate are also investigated. Lastly, by adopting the third-moment method, a cover cracking time probabilistic analysis is conducted.

## 2. Corrosion-Induced Cracking Model and Its Verification

Concrete is a heterogeneous material that inevitably contains randomly distributed initial defects with assorted mechanical behaviors. Those defects are attributable to the coactions of a number of factors such as overloading, shrinkage caused by restraint, weather (extremely dry, cold, or hot), poor workmanship, and settlement of concrete [36]. Precisely, the randomness associated to the size and the location of the initial defects will cause the emergence of variability for relative structures’ behaviors during service life, which are directly related to the safety of structures [20,37]. Therefore, in order to characterize the random responses of the relevant structure, several approaches were developed in the preceding decades, aiming to identify and evaluate the size and location of the initial defects. To date, a number of approaches have been successfully developed and implemented based on different methods, including infrared thermography, ground penetrating radar and radiographic methods, etc. [38].

### 2.1. Corrosion-Induced Cracking Model

In this section, the corrosion-induced cracking model is developed. In detail, this model is mainly involved in four aspects, including the determination of threshold pressure, the corrosion rate, weight loss and corrosion current density for steel bars in concrete. The development of the corrosion-induced cracking model is described as follows:

#### 2.1.1. The Threshold Pressure of the Initial Crack and Cover Crack

At the interface between the reinforced bar and concrete, the emanation of the initial defect is mainly attributed to the concrete settlement [22]. In this work, for these initial defects, a semi-elliptical crack existed in a thick-walled cylinder is adopted for modeling such an initial defect [32,33]. Additionally, in order to ensure the accuracy of relative analysis, the effects caused by these initial defects on the growth process of the corrosion-induced crack are also considered. In detail, Figure 1 shows an idealized section of a concrete cylinder with an initial defect which is assumed to be a three-dimensional semi-ellipsoid and can be simplified for plane problems. Specifically, in Figure 1, *c* denotes the half length of the semi-elliptical crack; *a* denotes the radial depth of the initial defect through the concrete cover; *R* denotes the inner steel radius; *C* denotes the concrete cover depth; the angle *β* is utilized to describe the position around the semi-elliptical crack varies between 0 ≤ *β* ≤ π, and *A* and *B* denote the deepest points of the crack in vertical and horizontal directions, respectively.

The theory of fracture mechanics is first introduced succinctly here in order to investigate the corrosion-induced cracking process in RC structures incorporating the effects of initial defects. In order to develop the model for predicting the time of corrosion-induced cracking on concrete cover, a model proposed by Liu and Weyers [6] is utilized, and relative parameters such as stress intensity factors are obtained from the literature [32,33]:

For the deepest of A:(1)$$K_{\mathrm{IA}}=pF_{A}\sqrt{\pi a/Q}$$

For the deepest of B:(2)$$K_{\mathrm{IB}}=pF_{B}\sqrt{\pi a/Q}$$
where *p* denotes the uniform internal pressure on the inner surface, and *F_A_* and *F**_B_* are the boundary correction factor of the initial defect which can be expressed as follows:(3)$$F_{A}=\frac{\sqrt{2Q}}{\pi }\left(G_{0}M_{1A}+G_{1}M_{2A}+G_{2}M_{3A}+G_{3}\right)$$
(4)$$F_{B}=\frac{2\sqrt{Q}}{\pi }\left(G_{0}M_{1B}+G_{4}M_{2B}+G_{5}M_{3B}+G_{6}\right)$$
where
(5)$$\begin{array}{l} G_{0}=\frac{F}{\left(R/C\right)u}+E\\ G_{1}=-\frac{F\ln u}{2{\left(a/C\right)}^{3/2}u^{1/2}}+\frac{F\ln\left(R/C\right)}{2{\left(a/C\right)}^{3/2}u^{1/2}}+\frac{F}{\left(a/C\right)\left(R/C\right)}+\frac{2}{3}E\\ G_{2}=-\frac{F\ln u}{{\left(a/C\right)}^{2}}+\frac{F}{\left(a/C\right)\left(R/C\right)}+\frac{F\ln\left(R/C\right)}{{\left(a/C\right)}^{2}}+\frac{1}{2}E\\ G_{3}=-\frac{Fw}{2{\left(a/C\right)}^{1/2}u^{3/2}}+\frac{F}{u\left(R/C\right)}-\frac{F\ln\left(R/C\right)}{2{\left(a/C\right)}^{1/2}u^{3/2}}+2E \end{array}$$
(6)$$\begin{array}{l} E=1+\frac{{\left(R/C\right)}^{2}}{2\left(R/C\right)+1}\\ F=\frac{{\left(R/C\right)}^{2}{\left(R/C+1\right)}^{2}}{2\left(R/C\right)+1}\\ w=\ln\left[u+\left(a/C\right)+2u^{1/2}{\left(a/C\right)}^{1/2}\right]\\ u=\left(R/C\right)+\left(a/C\right) \end{array}$$

The parameters *M_iA_* were obtained by
(7)$$\begin{array}{l} M_{1A}=\frac{2\pi }{\sqrt{2Q}}\left(-Y_{0}+3Y_{1}\right)-\frac{24}{5}\\ M_{2A}=3\\ M_{3A}=\frac{6\pi }{\sqrt{2Q}}\left(Y_{0}-2Y_{1}\right)+\frac{8}{5} \end{array}$$

The geometry correction factors, *Y*_0_ and *Y*_1_ are expressed as
(8)$$\begin{array}{l} Y_{0}=A_{0}+A_{1}\left(\frac{a}{C}\right)+A_{2}{\left(\frac{a}{C}\right)}^{2}+A_{3}{\left(\frac{a}{C}\right)}^{4}\\ Y_{1}=B_{0}+B_{1}\left(\frac{a}{C}\right)+B_{2}{\left(\frac{a}{C}\right)}^{2}+B_{3}{\left(\frac{a}{C}\right)}^{4} \end{array}$$
where *A_i_* and *B_i_* are gained by
(9)A0=0.07e[−5.051(ac)]+1.044A1=0.665e[−3.393(ac)]−0.433A2=1.16e[−3.386(ac)]+0.711A3=1.46e[−4.165(ac)]−0.179
(10)B0=−2.16e[−0.035(ac)]+2.825B1=0.265e[−5.574(ac)]−0.225B2=0.753e[−4.0251(ac)]+0.307B3=−1.284e[0.0791(ac)]+1.398

*Q* is the elliptical crack shape factor:(11)$$Q=1+1.464{\left(\frac{a}{c}\right)}^{1.65}$$
where the value of the stress intensity factor reaches the fracture toughness, the crack starts to propagate. Double *K* fracture criterion for mode I crack of concrete can be represented as follows [39,40]:
(12)$$\begin{array}{l} K_{I}<K_{\mathrm{Ic}}^{\mathrm{ini}}\;\mathrm{no}\;\mathrm{crack}\;\mathrm{propagation};\\ K_{\mathrm{Ic}}^{\mathrm{ini}}<K_{I}<K_{\mathrm{Ic}}^{\mathrm{un}}\;\mathrm{steady}\;\mathrm{crack}\;\mathrm{propagation};\\ K_{I}>K_{\mathrm{Ic}}^{\mathrm{un}}\;\mathrm{unsteady}\;\mathrm{crack}\;\mathrm{propagation}; \end{array}$$
where *K*_I_ denotes the stress intensity factor appeared in Equation (1); $K_{\mathrm{Ic}}^{\mathrm{ini}}$ and $K_{\mathrm{Ic}}^{\mathrm{un}}$ denote the initial fracture and the unstable fracture toughness, respectively (named double-K fracture parameters).

Generally, the double-K fracture parameters are utilized to analyze concrete problems involved with the crack initiation and growth [41], and a variety of studies associated to experimental observations and analytical methods attempting to determine these parameters can be traced [41,42,43]. However, due to absence of the consideration on the coupling of size and boundary effects, utilization of the double-K fracture parameters is limited in the corrosion-induced cracking process. Therefore, Zhang et al. [22] adopted a correction method by treating the reinforced concrete thick wall cylinder as a three-point bending notched beam. The relationship between the modified stress intensity factor and the experimental data is expressed as [39]:(13)$$K_{\mathrm{Ics}}=K_{\mathrm{Ic}}\sqrt{\frac{C}{h}}{\left(\frac{V}{2R\cdot C}\right)}^{1/\alpha^{\prime }}(h\leq 2m)$$
where *h*, *V*, and *K*_Ic_ are the height, volume and fracture toughness of a three-point bending beam, respectively; $\alpha^{\prime }$ is the Weibull modulus related to the variation of experimental results, which can be obtained by
(14)$$\alpha^{\prime }=\frac{\pi }{\sqrt{6.5C_{V}}}\;(0.013\leq C_{V}\leq 0.230)\;$$

Based on the theory of fracture mechanics, the stress intensity of the initial defect will increase with the growth of the rust expansion force generated by the expansion of steel corrosion products. When the increase in the stress intensity factor is equal to the initial fracture toughness of the concrete material, the location of the initial defects first started to crack, namely the first phase of the concrete crack, crack length increases as the corrosive force continues to increase, eventually forming a well versed in the crack of the concrete surface; this phase is the concrete cover cracking criterion completely. For the semi-elliptical flaw, the equilibrium *a*/2*c* always increases to a limiting value of 0.36 [44]. This results from a variation in the stress intensity factor *K*_I_ along the surface of the ellipse. When *β* = 90°, *K*_I_ is maximized, but is smallest when *β* = 0°. Hence, the crack will grow fastest when *β* = 90°.

It can be known from the above discussion, when calculating the crack propagation of concrete initial defects, two critical points should be considered, viz. the initial defects cracking and the complete cracking of concrete cover depth. It is assumed that the crack stress intensity factors of these two critical points are equal to the double K fracture parameters, so, we can obtain the initial threshold expansion pressure *p*^ini^ and the threshold cracking pressure *p*^un^ at the initial defect points A and B by substituting Equation (1) and Equation (2). Since the crack development of the initial defect at point B is along the reinforcement bar, which has little effect on the crack expansion of the initial defect at point A, considering the crack state of the initial defect at point A as the criterion for judging the crack penetration of the concrete cover depth.

The initial threshold pressure *p*^ini^ is as follows:(15)$$P^{\mathrm{ini}}=\frac{K_{\mathrm{Ics}}^{\mathrm{ini}}}{F_{A}\sqrt{\pi a/Q}}\;$$

Hence, the expression for the threshold cracking pressure $P^{\mathrm{un}}$ is given as:(16)$$P^{\mathrm{un}}=\frac{K_{\mathrm{Ics}}^{\mathrm{un}}}{F_{A}\cdot \sqrt{\pi (a+\Delta a^{*})/Q}}\;$$
where Δa^*^ denotes the length of fracture zone and can be obtained by following equation [19]:(17)$$\Delta a^{*}=\frac{1}{2\pi }{\left(\frac{K_{\mathrm{Ics}}^{\mathrm{un}}}{f_{t}}\right)}^{2}$$
where $f_{t}$ is the tensile strength of concrete. 

#### 2.1.2. The Corrosion Rate of Steel Bar 

For an initial unrestrained RC specimen with the bottom clear cover *C* and original reinforcing bar *R*, the thick-walled concrete cylinder is shown in Figure 2. The original radius of the steel bar is *R*, the radius of steel bar after corrosion is *a*_1_, the radial loss of steel is *R-a*_1_, and the combined radius of un-corroded steel plus free-expansive corrosion products is *a*_2_. Based on the amount of theoretical analysis and experimental study, it is verified that there are a lot of porous zones around the interface between the reinforcement bar and concrete, which affects the cracking time of concrete cover depth. For the sake of simplicity, the porous zone is assumed to be uniform and its thickness is indicated by *d*_0_, assumed to be 12.5 mm as adopted by Liu and Weyers [6], Bhargava et al. [45] and Kotes [46]. The corrosion products must first fill this porous zone before their volume expansion starts to create uniform radial inner pressure *p* around the surrounding concrete, due to which the concrete gets an internal radial displacement *σ*_con_, therefore:(18)$$\sigma_{\mathrm{con}}=\frac{p}{E_{\mathrm{ef}}}\cdot \frac{R{\left(R+C\right)}^{2}\left(1+\upsilon_{c}\right)+R^{3}\left(1-\upsilon_{c}\right)}{2RC+C^{2}}$$
where *u*_c_ denotes the Poisson’s ratio; *E*_ef_ denotes the effective modulus of elasticity for concrete cover, which can be obtained as:(19)$$E_{ef}=\frac{E_{c}}{1.0+\varphi }$$
where *E*_c_ denotes elastic modulus; *j* denotes creep coefficient of the concrete cover. The value for *j* as per the reference is 2.0 [6].

Based on the geometric condition shown in Figure 2, the corrosion-induced loss of volume per meter on the longitudinal direction of steel can be written as:(20)$$\Delta V_{\mathrm{steel}}=\pi (R^{2}-a_{1}{}^{2})$$

Therefore, the corrosion rate denoted as *ρ* can be written as follows:(21)$$\rho =\frac{\Delta V_{\mathrm{steel}}}{V_{\mathrm{steel}}}$$

Additionally, the volume of the porous zone per meter on the longitudinal direction can be expressed as:(22)$$\Delta V_{p}=2\pi Rd_{0}$$

In addition, the volume of concrete per meter on the longitudinal direction caused by the radial displacement *σ*_con_ can be determined as:(23)$$\Delta V_{\mathrm{con}}=2\pi (R+d_{0})\sigma_{\mathrm{con}}$$

For the sake of simplicity, the volume of crack per meter on the longitudinal direction after concrete cover cracking can be estimated by [47,48]:(24)$$\begin{array}{l} \Delta V_{\mathrm{crack}}=\frac{1}{3}{\sum \omega }_{i}ac\;\\ {\sum \omega }_{i}=2\pi R(n-1)(1-\sqrt{1-\rho })\; \end{array}$$
where *n* is the volume ratio between the corrosion products and the basic steel. The expression is given as follows:(25)$$n=\Delta V_{rust}/\Delta V_{steel}$$

Generally, when concrete cover begins to crack, corrosion-induced product penetration will occur in both the porous zone and cracks. Hence, the Δ*V*_rust_, which denotes the total corrosion volume per meter on the longitudinal direction, consists of four parts: (26)$$\Delta V_{\mathrm{rust}}=\Delta V_{p}+\Delta V_{\mathrm{con}}+\Delta V_{\mathrm{crack}}+\Delta V_{\mathrm{steel}}$$

Combining Equations (19) to (25) obtains a relationship as follows:(27)$$\pi \eta \rho R^{2}=\frac{2}{3}\pi (n-1)acR(1-\sqrt{1-\rho })+2\pi Rd_{0}+2\pi (R+d_{0})\sigma_{\mathrm{con}}+\pi \rho R^{2}$$

Solving Equation (21), the corrosion rate *ρ* can be obtained:
(28)$$\rho =\frac{-b_{1}\pm \sqrt{b_{1}^{2}-4a_{3}c_{1}}}{2a_{3}}$$

Hence, one can see that there is a strong correlation between the corrosion rate *ρ* and the shape of the initial defect.

#### 2.1.3. Steel Bar Weight Loss Calculation

When concrete cover cracks, the loss of steel weight can be expressed as follows:(29)$$M_{\mathrm{loss}}=\rho_{\mathrm{cr}}M_{s}=0.0616\rho_{\mathrm{cr}}D^{2}$$
where *M*_s_ is the original steel weight (g); *ρ*_cr_ is the corrosion rate of steel bar when cover cracks (%); *D* is the diameter of steel bar (mm).

The relationship among time, corrosion rate of the reinforcing bar and the loss of reinforcement weight is established based on Faraday’s law of corrosion and can be expressed as follows:(30)$$M_{\mathrm{loss}}=\frac{MI_{\mathrm{corr}}}{zF}t$$
where *M*_loss_ is the loss of steel wight (g); *z* is lon valence; *I*_corr_ is corrosion current (A); *M* is the molecular mass of iron (g); *t* is corrosion time (s); *F* is Faraday’s constant (96500C).

#### 2.1.4. Corrosion Current Density of Steel Bars in Concrete

During the cracking process for concrete cover, the corrosion current density of the reinforcing bar is regarded as one of the determinant factors on relative behaviors. In past decades, a number of researchers have conducted several in-depth researches on this topic. In this work, the corrosion current density utilized is determined by Lu et al. [29], and the expression is given as:(31)$$i_{\mathrm{corr}1}\left(t\right)=\frac{1}{\sqrt[{3}]{1+t}}\exp\left[1.23+0.618\ln C_{t}-\frac{3034}{T\cdot \left(2.5+RH\right)}-5\times 10^{-3}\rho^{0}\right]$$
where *i*_corr_ denotes corrosion current density (μA/cm^2^); *C_t_* denotes concrete chloride content on the surface of reinforcement (kg/m^3^); *T* is the temperature (K); *RH* is relative humidity; *t* is corrosion duration (years); *ρ*^0^ is concrete resistivity (kohm.cm).

The above formula is mainly used for steel corrosion in the natural corrosion environment, while the corrosion current density in the steel corrosion accelerated by electrification is artificially set and a known parameter.

Past studies [3] have shown that concrete cover cracking time is closely related to cover depth, the diameter of steel reinforcement, the strength of concrete, concrete water–cement (*w/c*) ratio and other factors. Concrete *w/c* ratio has a great influence on concrete strength; the larger the concrete *w/c* ratio is, the lower the concrete strength is. In this paper, the concrete *w/c* ratio is selected as one of the main factors affecting cover cracking. Similarly, cover depth also has an important effect on cover cracking time; the thicker the cover, the longer it takes for the crack to reach the concrete surface, and the longer it takes for the corrosion products to fill the pores in the concrete [17]. Based on the Vu and Stewart model [27], the formula of impact factor is established, considering the influence of the *w/c* ratio and cover depth on cover cracking time and can be expressed as: (32)$$f_{\mathrm{conc}}=k\frac{{\left(1-w/c\right)}^{2}}{C}$$

Substituting Equation (32) into (31), the following relationship is obtained:(33)$$i_{\mathrm{corr}1}\left(t\right)=k\frac{{\left(1-w/c\right)}^{2}}{C}\cdot \frac{1}{\sqrt[{3}]{1+t}}\exp\left[1.23+0.618\ln C_{t}-\frac{3034}{T\cdot \left(2.5+RH\right)}-5\times 10^{-3}\rho^{0}\right]$$
where *k* is the adjustment coefficient. 

By conducting regression analysis on the experimental results proposed by Liu and Weyers [26], a modified formula of corrosion current density can be obtained:(34)$$i_{\mathrm{corr}1}\left(t\right)=62.69\frac{{\left(1-w/c\right)}^{2}}{\sqrt[{3}]{1+t}\cdot C}\cdot \exp\left[1.23+0.618\ln C_{t}-\frac{3034}{T\cdot \left(2.5+RH\right)}-5\times 10^{-3}\rho^{0}\right]$$

#### 2.1.5. Cracking Time Model

In the past, cover cracking time was mainly studied in the natural corrosion environment and the electrified acceleration environment; in the natural corrosion environment, the corrosion current density will change with time, and in the electrified acceleration environment, the corrosion current density is a constant value, so, this paper needs to consider cover cracking time in these environments.

For reinforced concrete structures in naturally corrosive environments, combining Equations (29) to (30) and Equation (34), the following relationship is obtained:(35)$$M_{\mathrm{loss}}=0.0616\rho_{\mathrm{cr}}D^{2}=\frac{M}{zF}\pi D\int_{0}^{t}i_{\mathrm{corr}1}dt$$

By integrating Equation (35), the formula of concrete cover cracking time is obtained:(36)$$t_{\mathrm{cr}}={\left[\frac{0.285\rho_{\mathrm{cr}}DC{\left(1-w/c\right)}^{1.64}}{H^{*}}+1\right]}^{1.5}-1$$
where *H** can be expressed as:(37)$$H^{*}=\exp\left[1.23+0.618\ln C_{t}-\frac{3034}{T\cdot \left(2.5+RH\right)}-5\times 10^{-3}\rho^{0}\right]$$

For reinforced concrete structures in naturally corrosive environments, based on the Wang [34] model, the cover cracking time can be expressed as follows:(38)$$t_{\mathrm{cr}}=78.3\frac{\rho_{\mathrm{cr}}D}{i_{\mathrm{corr}}}=78.3\frac{D\left(-b_{1}\pm \sqrt{b_{1}^{2}-4a_{3}c_{1}}\right)}{i_{\mathrm{corr}}\cdot 2a_{3}}$$
where *a*_3_*, b*_1_*, c*_1_ is the combination coefficient, which can be obtained by Equation (27).

### 2.2. Experimental Verification of Cracking Model

In order to verify the rationality and applicability of the above model in this study, five-year naturally exposed experiments, short time indoor tests and corroded concrete members in service on the prediction of time to cracking conducted by Liu and Weyers [6], Wang [34] and Shi [35] were used. The determination process of the relative parameters involved with the present model are explained as below:

Initially, according to the experimental results conducted by Wu et al. [49], for concrete, the stable values of $K_{\mathrm{Ic}}^{\mathrm{ini}}$ and $K_{\mathrm{Ic}}^{\mathrm{un}}$ are selected as 1.034 MPa·m^1/2^ and 2.072 MPa·m^1/2^. Secondly, the value of the corresponding coefficients variation are determined as 0.061 and 0.073, respectively. In addition, other parameters for the proposed model are chosen as the same as the results adopted by Liu and Weyers [6], Wang [34] and Shi [35].

#### 2.2.1. Compared to the Experimental Data of Liu and Weyers

The experimental data of slabs in reference [6] is listed in Table 1, and the comparison results of different models with the experimental data are listed in Table 2.

It was shown that in Table 2, there is a relatively good coincidence between the prediction by the proposed model and the experimental results on the cover cracking time. Meanwhile, the predictions by the proposed model are comparatively improved when compared to that yielded by the models of Liu and Weyers [6] and Zhang et al. [50]. Moreover, the proposed model is also capable of evaluating the threshold pressure and corrosion rate, therefore, it is verified that the proposed model is able to predict the cover cracking time for both indoor and outdoor specimens.

#### 2.2.2. Compared to the Acceleration Experimental Data of Wang

The acceleration experimental data of specimens in reference [34] are listed in Table 3, and the comparison results of different models with experimental data are listed in Table 4.

It can be seen from Table 4, there is a good matching between the predictions of the proposed model and the experimental results achieved on the cover cracking time of accelerate corrosion specimens. In addition, when compared to the predictions generated by the models of Wang [34] and Maaddawy et al. [16], it is shown that the accuracy of the predictions by the proposed model is improved.

#### 2.2.3. Compared to the Actual Measuring Data of Shi

The actual measuring data of members in reference [35] and the comparison results are listed in Table 5.

Table 5 indicates the comparison results between the predictions by the proposed model and measuring results on the cover cracking time. It is revealed that the prediction is in good agreement with the actual measuring results. Therefore, it is proved that the proposed model can well predict the time to cover cracking of concrete structures in service.

## 3. Influences of Various Parameters on Threshold Pressure, Corrosion Rate and Cracking Time

In order to reflect the effectiveness of parameter influence, the parameters in the model need to be standardized. Based on engineering experience, the settlement height of concrete is about 1% of the concrete height; when the specimen height is about 200 mm, the length of initial defects is defined as 2 mm [20]. According to the selection standard of critical chloride concentration [51], the critical chloride ion concentration is 0.2%, assuming the mass of concrete is 2400 kg/m^3^, and the total chloride content (reinforcement surface) is calculated to be 4.8 kg/m^3^. In addition, the standard values for other variables need to be determined, and the value of other basic variables is listed in Table 6. 

### 3.1. Influence of the Initial Defect Length

The initial defect length (*a*) is set to increase from 2 to 6 mm at 1 mm, and the corresponding initial defect size (*a/c*) is 0.1, 0.15, 0.2, 0.25, and 0.3, respectively. When the concrete cover cracks, the initial crack growth limits (*a/c*) are 0.64, 0.72 and 0.8. The calculation results of threshold pressure, corrosion rate of rebar and cover cracking time obtained by substituting the calculation parameters into the model are shown in Figure 3.

Figure 3 shows that the initial defect length (*a*) has a similar variation trend to the threshold pressure, corrosion rate of rebar and the cover cracking time, all of which gradually decrease with the increase in the initial defect length. This is because the increase in the initial defect length indicates that the on-site pouring and curing concrete are not in place, or that the concrete construction technology needs to be improved, and concrete density becomes worse, leading to the advance of the cover cracking time. When *a/c* = 0.72, the initial defect length increases from 2 mm to 7 mm, and the corresponding reduction rates of the threshold cracking pressure, corrosion rate and cover cracking time are 19.66%, 3.54% and 4.12%, respectively, indicating that the initial defect length has a greater impact on the threshold cracking pressure, followed by the cover cracking time. The initial defects cracks, or concrete cover cracks, the initial threshold pressure, the threshold cracking pressure, corrosion rate of rebar and cover cracking time all increase with the increase in the initial defect size (*a/c*). This is because the greater the ratio of the initial defect size, the smaller the initial defects area of the semi-ellipse is, indicating that the influence of the initial defect size on concrete cover cracking is reduced. In general, the initial defect size will change the cracking behavior and the failure mechanics mode of concrete cover cracking and affect the critical indexes such as concrete cover cracking time. At present, the relevant experimental and theoretical research of the initial defect size on the concrete cover cracking process is still blank.

### 3.2. Influence of the Expansion Rates of Corrosion Products

Based on previous research results, the range of the expansion rates of corrosion products is 2–4 times, which is set to increase from 2 to 4 at 0.5. The calculation results of the corrosion rate and cover cracking time obtained by substituting the calculation parameters into the model are shown in Figure 4.

Figure 4 shows that the expansion rates of corrosion products have a greater impact on the corrosion rate of rebar and cover cracking time, as the expansion rates of corrosion products increase from 2.0 to 4.0, the reduction rates of the corrosion rate and cover cracking time is 199.33% and 226.32%, respectively, and the difference between the two is not big; there is a one-to-one correspondence. This is because the limited pore area inside the concrete cannot satisfy the expansion of the corrosion product volume greatly, in the case of a low rate of reinforcement corrosion rust had a great expansion pressure caused by coating concrete cover cracking in advance, so the selection of reasonable expansion rates of corrosion products for predicting the cover cracking time is crucial.

### 3.3. Influence of the Thickness of the Porous Zone

Currently, the thickness of the porous zone at the interface was selected by referring to the data provided by other researchers, which is set to increase from 5 to 25 μm at 5 μm. The calculation results of the threshold pressure, corrosion rate of the rebar and cover cracking time obtained by substituting the calculation parameters into the model are shown in Figure 5.

The results of Figure 5 show as the thickness of the porous zone increases from 5 to 25 μm, the growth rates of the corresponding threshold pressure, corrosion rate of the rebar and cover cracking time are 0%, 203.61% and 226.32%, respectively, which indicates that the thickness of the porous zone has a relatively large effect on the corrosion rate of the rebar and cover cracking time but has little effect on the initial threshold pressure and the threshold cracking pressure. This is because the pressure is mainly sensitive to the initial defect size, which will affect the fracture expansion trend, while the thickness of the porous zone will not affect the fracture development. However, the increase in the thickness of the porous zone improves the porosity of concrete and provides more storage space for corrosion products, which leads to the increase in rust expansion and cover cracking time. 

### 3.4. Influence of Cover Depth

The cover depth is set to increase from 20 to 65 mm at 5 mm; the calculation results of the threshold pressure, corrosion rate of rebar and cover cracking time are obtained by substituting the calculation parameters into the model, as shown in Figure 6.

Results of Figure 6 show that as cover depth increases from 20 mm to 65 mm, the growth rates of the threshold cracking pressure, corrosion rate of the rebar and cover cracking time are 38.65%, 10.54% and 326.9%, respectively, which indicates that cover depth has the largest impact on cover cracking time, and a relatively small impact on the threshold pressure and corrosion rate of the rebar. Therefore, for some structures with a long design life and commemorative significance, such as bridges in coastal areas and museums in various regions, the service life of the structures can be improved by increasing the cover depth.

### 3.5. Influence of w/c Ratio and Chloride Content

The concrete water–cement (*w/c*) ratio is set to increase from 0.3 to 0.7 at 0.1, and chloride content (reinforcement surface) is set to increase from 4.8 to12.0 kg/m^3^ at 1.2 kg/m^3^. The calculation results of cover cracking time obtained by substituting the calculation parameters into the model are shown in Figure 7.

Results of Figure 7 show that as the *w/c* ratio increases from 0.3 to 0.7, the cover cracking time decreases by 244.7% significantly, indicating that the *w/c* ratio has a great influence on cover cracking time. This is because the *w/c* ratio will affect the tensile strength of concrete, so the corrosion rate of the reinforcing bar increases, and the corresponding cover cracking time will be shortened. Moreover, the *w/c* ratio has an indirect effect on the threshold pressure and corrosion rate of the rebar by affecting the porosity and tensile strength of concrete. As the concrete chloride content increased from 4.8 kg/m^3^ to 12 kg /m^3^, the corresponding growth rate of cover cracking time was 85.48% significantly. This is because in the steel bar corrosion in the event of a reaction, chloride ions have played a catalytic role but are not consumed themselves; as concrete chloride content increased, the process of steel corrosion was accelerated virtually, resulting in cover cracking time greatly in advance.

## 4. The Cover Cracking Time Probabilistic Analysis

### 4.1. Probability Model Based on the Third Moment (TM) Method

In fact, the randomness is one of the essential characteristics for concrete material and is closely related to structure safety. For the problem of estimating the cover cracking time in actual projects, from a practical point of view, approaches based on the deterministic method are limited. Hence, in order to reflect the real response on the cover cracking time, it is necessary to adopt the random probability theory on relevant studies. In this work, the TM method proposed by Stanish et al. [52], Zhang et al. [53] and Zhao et al. [54,55] is adopted to calculate the reliability and probability of failure for reinforced concrete structures. 

For a reinforced concrete structures, the service life (lifetime) is defined as the period of time that meets or exceeds the initial requirements. Normally, a performance-based assessment criterion needs to be established firstly for determination of the service life of reinforced concrete structure. The general form for such criterion of limit state function is given as follows:(39)G(L,S,t)=L(t)−S(t)
where *S*(*t*) is the action (load) or its effect at time *t* and *L*(*t*) is the acceptable limit (resistance) for the action or its effect. 

Based on Equation (47), the probability of concrete (structural) failure, *P_f_* can be determined by
(40)$$P_{f}\left(t\right)=P\left[G\left(L,S,t\right)\leq 0\right]=P\left[S\left(t\right)\geq L\left(t\right)\right]$$

It is noted that for concrete, once the *K*_I_ (stress intensity factor) is larger than *K*_Ic_ (fracture toughness), concrete cover is not able to sustain the crack growth. Therefore, the *t_cr_* implies the critical time for cracks appearing in the concrete cover (gained by the value of *K*_Ic_ of concrete cover), *t* represents the changed time when corrosion products gradually accumulate. Based on the analytical solution, it can be found that the governing parameters (*C, d, T, RH, E_c_*, *Cl*, *w/c*, *K*_Ic_) are considered as random variables. Therefore, the probability of concrete failure can be determined from Equation (41) directly, where *t* replaces *S* and *t_cr_* replaces *L*, as follows:(41)$$p_{f,c}(t)=P\left[t\geq t_{cr}\right]$$

### 4.2. Verification of the Proposed Model

In this section, the efficiency and accuracy of the proposed model on the probability of corrosion-induced failure is verified with different cover cracking. Specifically, comparisons between the results of the proposed model and those obtained by traditional Monte Carlo (MC) simulations are conducted. In addition, the standard, mean, coefficient of variation (COV) and distribution type of several fundamental variables are obtained and listed in Table 6 and Table 7.

In detail, results yielded by the proposed model and MC simulations for cover cracking failure probability with sample sizes N = 10, 100, 1000 and 10,000 are listed in Figure 8. It is found that, when the sample size N is less than 1000, the prediction of failure probability of the MC simulation is unstable. Additionally, for the case of the sample size N = 10,000, the predictions generated by both the proposed model and the MC simulation are in good agreement. However, the TM method used in the proposed model is simpler than the MC simulation when dealing with the analysis of the probability of cover cracking.

### 4.3. Time-Dependent Probability Assessment of the Proposed Model

#### 4.3.1. Mean Value of Various Factors

By adopting the value of various factors (*C*, *d*, *T*, *RH*, *E_c_*, *Cl*, *w/c*, *K*_Ic_) in Table 7, the corresponding failure probability of cover cracking influenced by their mean values of these factors are illustrated in Figure 9.

It can be seen from in Figure 9 that the influence of the mean values of these factors shows a similar trend, and the failure probability of cover cracking also increases with the growth of time.

Cove depth, the diameter of steel bar and the water–cement ratio are the most important factors affecting cracking probability (Figure 9a,b,d). Moreover, it is found that there are also great effects on the probability of cracking for chloride content, temperature and relative humidity (Figure 9c,e,f). Specifically, in Figure 9a, it shows that as the value of *C* increases from 30 mm to 60 mm, the surface cracking time, which is defined as the time to reach a 50% cracking probability, decreases from approximately 0.8 years to 2.7 years. In Figure 9b, it shows that when *d* varies from 16 mm to 25 mm, it results in an approximately 1.2-year increase in the surface cracking time. In Figure 9d, the surface cracking time increases from 0.5 years to around 2.7 years as *w/c* changes from 0.35 to 0.65. In Figure 9c,e,f, an increase in *C_t_*, *T, RH* from 4.8 to 9.6 kg/m^3^, 283 to 313 K, 0.65 to 0.95 can lead to a reduction in the surface cracking time between 1.0 and 2.0 years, respectively.

As shown in Figure 9g,h, the probability curve of the elastic modulus *E_c_* and cracking fracture toughness *K*_Ic_ is similar to each other, and the results of the effects of two factors on the cover cracking probability are negligible.

The comparison results of effects of cover depth, chloride content and temperature are summarized in Figure 9a,c,e. As indicated in these figures, the calculation results of the two methods are very close, which shows that the third-order method can calculate the influence of these three factors on the cover failure probability more accurately.

#### 4.3.2. Coefficient of Variation

Due to the variation of the values for the above influencing factors, relative effects by coefficient of variation (COV) on the probability of cracking need to be examined. In detail, the relevant results are represented in Figure 10.

It can be seen from Figure 10 that the effects of COV for different factors on the failure probability of cover cracking shows a similar trend, and the failure probability of cover cracking also increases with the growth of time.

Specifically, it is concluded that the COV of cove depth and chloride content are the factors that affect cracking probability the most (Figure 10a,c). Moreover, the *w/c* ratio, relative humidity and cracking fracture toughness also have a great influence on the cracking probability (Figure 10d,f,h). In Figure 10a, an increase in the coefficient of variation (*C*) from 0.1 to 0.6 has no effect on cover cracking time. In Figure 10c, it shows that the cover cracking time, which emerges at a 0.2-year increase, increases as *Cl* changes from 0.1 to 0.7. In Figure 10d,f,h, an increase in the coefficient of variation of *d*, *w/c* ratio, *K*_Ic_ from 0.1 to 0.6, 0.05 to 0.3, 0.1 to 0.4 can lead to a increment in cover cracking time between 1.8 and 1.9 years, respectively.

As shown in Figure 10b,e,g, the probability curve of the COV for the diameter of the steel bar *d*, along with temperature *T* and the elastic modulus *E_c_* on the cover cracking probability are similar to each other, and the results show that the influence of these three factors on the cover cracking probability is negligible.

The comparison results of the effects of the coefficient of variation of cover depth, chloride content, diameter of steel bar and temperature are summarized in Figure 10a,b,c,e. The results show the calculation results of the two methods are very close, which shows that the third-order method can calculate the influence of these factors on the cover failure probability more accurately.

## 5. Conclusions

In this study, a corrosion-induced cracking model based on the theory of fracture mechanics was proposed. Specifically, in this model, the three-dimensional initial defect, modified corrosion rate of the reinforcing bar in a natural environment and the acceleration to power corrosion were taken into account. In addition, the influence of relative parameters on several critical corrosion-induced crack indices were also analyzed. The proposed model is capable of quantitatively evaluating the risk of cracking for RC structures and predicting their service life. The following conclusions can be drawn:
(1)The predictions yielded by the proposed model are in close agreement with the experimental results obtained from published data and achieve better precision when compared with other existing models.(2)Through sensitive analysis of relative parameters in the cracking model, it was found that critical indices including threshold pressure, corrosion rate and cracking time decrease slowly when the initial defect size increases from 2 mm to 8 mm. Besides, both the volume expansion ratio of corrosion products and the thickness of the porous zone had great effects on the critical indices of corrosion rate and cracking time. It is found that the values of these two critical indices decline sharply with the increase in volume expansion ration from 2.0 to 4.0 and the decrease in the thickness of the porous zone from 25 μm to 5 μm, respectively. It is shown that other parameters including cover depth, *w*/*c* ratio and chloride content were only influential to the cracking time.(3)The results of probabilistic analysis for cover cracking time revealed that there are certain effects on the failure probability of cracking caused by both the mean values and COV of basic variables. The cracking risk of RC structures was found to grow with the increase in the mean value for chloride content, temperature, relative humidity and *w*/*c* ratio, and decreased with the increase in cover depth and the steel bar diameter. Among the *w*/*c* ratio, cover depth and steel bar dimeter, the cracking time was more sensitive to the *w*/*c* ratio than the other two variables.(4)Compared with traditional methods such as the MC simulation, the proposed TM method was able to obtain higher efficiency and greater simplicity in the stochastic analysis of corrosion-induced cracking problems.


## Figures and Tables

**Figure 1 materials-14-01440-f001:**
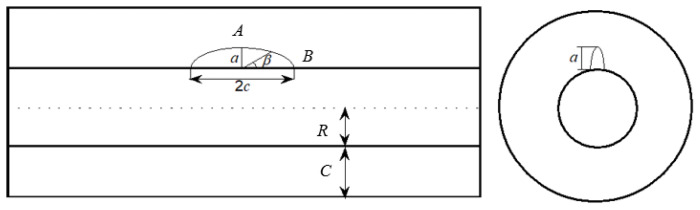
Schematic diagram of semi-elliptic initial defect: *c*—half length of the semi-elliptical crack; *a*—radial depth of the initial defect; *R*—inner steel radius; *C*—concrete cover depth; *β*—position angle; *A*—deepest points of the crack in vertical direction; *B*—deepest points of the crack in horizontal directions.

**Figure 2 materials-14-01440-f002:**
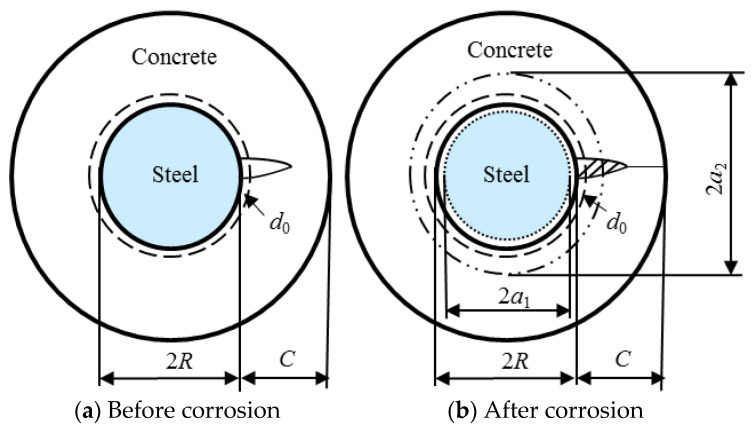
Calculation model of corrosion rate: (**a**) before corrosion; (**b**) after corrosion; *R*—original radius of steel bar; *a*_1_—radius of steel bar after corrosion; *a*_2_—combined radius of un-corroded steel plus free-expansive corrosion product; *d*_0_—the thickness of porous zone.

**Figure 3 materials-14-01440-f003:**
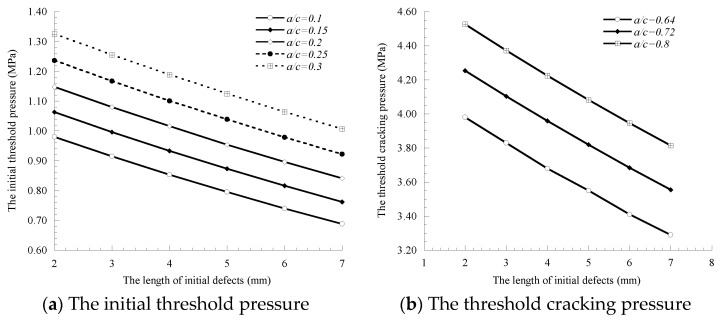
Influence of fine crack on threshold pressure, corrosion rate and cracking time.

**Figure 4 materials-14-01440-f004:**
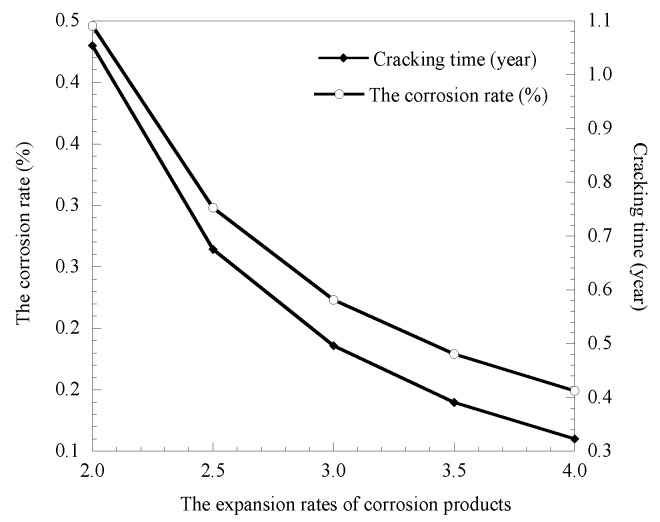
Relationship between corrosion products expansion rates and corrosion rate as well as cover cracking time.

**Figure 5 materials-14-01440-f005:**
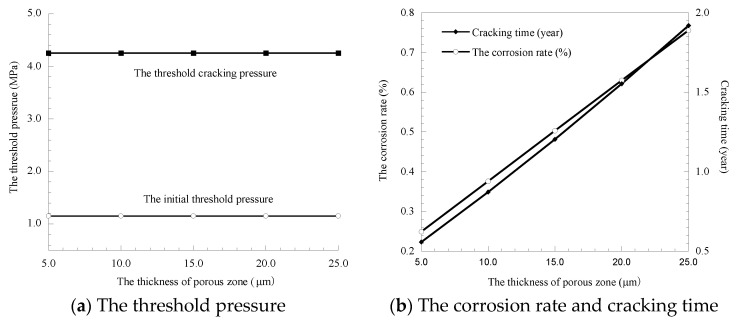
Influence of the thickness of the porous zone.

**Figure 6 materials-14-01440-f006:**
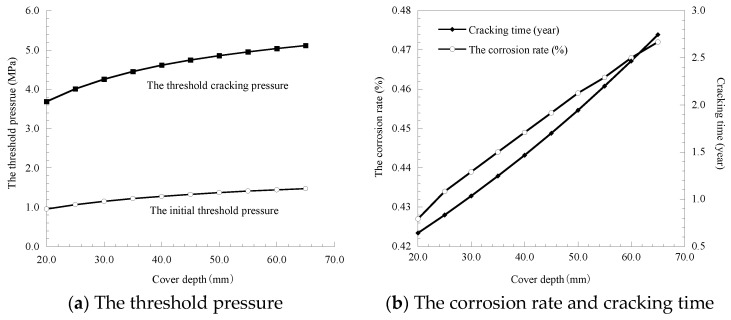
Influence of the cover depth on threshold pressure, corrosion rate and cracking time.

**Figure 7 materials-14-01440-f007:**
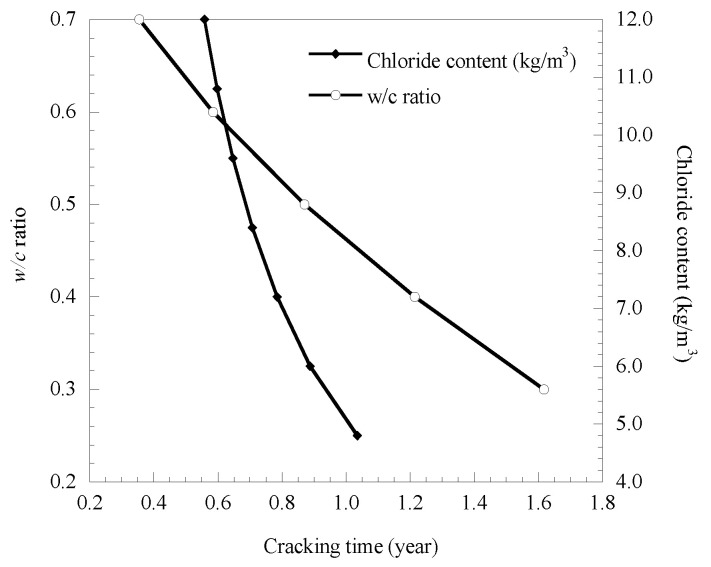
Influence of *w/c* ratio and chloride content on cover cracking time.

**Figure 8 materials-14-01440-f008:**
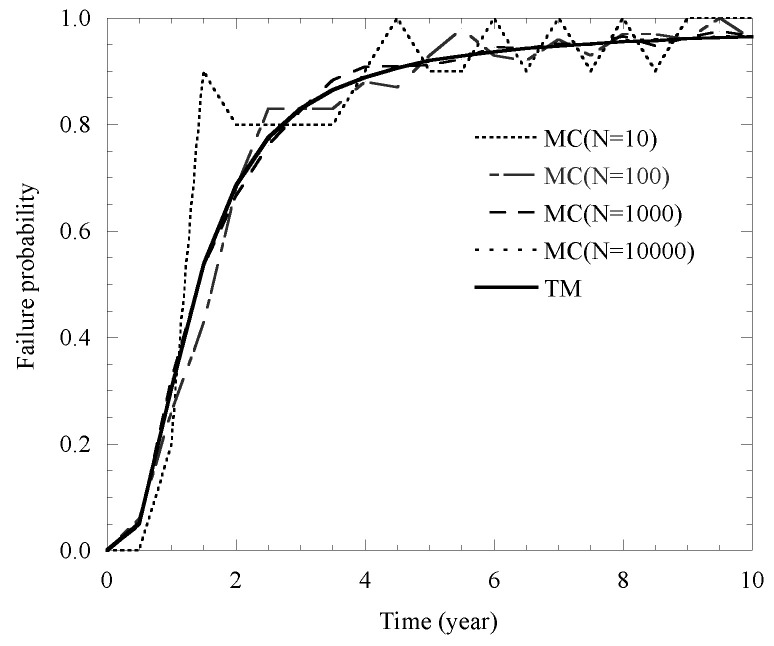
Comparison of TM and MC on cover cracking probability.

**Figure 9 materials-14-01440-f009:**
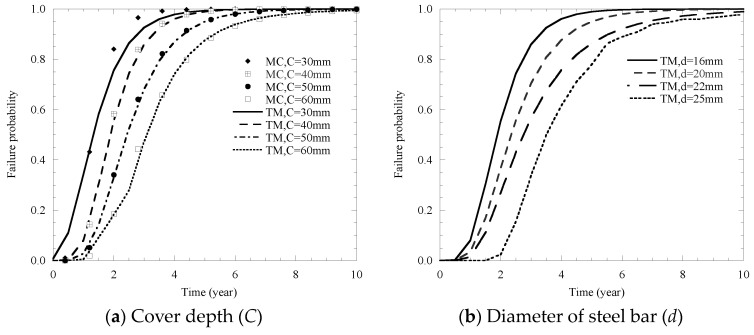
Influence of the mean values of different factors on the failure probability of cover cracking.

**Figure 10 materials-14-01440-f010:**
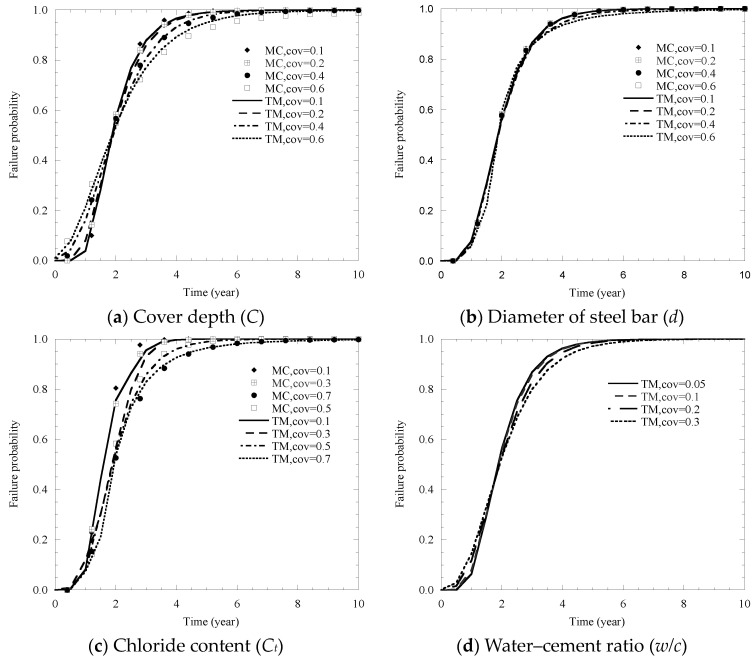
Influence of the coefficient of variation of different factors on failure probability of cover cracking.

**Table 1 materials-14-01440-t001:** The value of basic parameters. *C_t_*: chloride content; *w/c*: water–cement ratio; *T*: temperature; *E*_c_: elastic modulus; *vc:* Poisson’s ratio; *f*_t_: tensile strength.

Specimen Designation	2R/mm	C/mm	*w/c* Ratio	*C_t_*/kg/m^3^	*T*/K	*E*_c_/MPa	*ν* _c_	*ϕ*	*f*_t_/MPa	Exposure Period/yr
S1	16	48	0.43	4.92	295	27,000	0.18	2.0	3.3	1.84
S2	16	70	0.43	4.92	295	27,000	0.18	2.0	3.3	3.54
S3	16	27	0.45	6.02	295	27,000	0.18	2.0	3.3	0.72
S4	12.7	52	0.43	4.92	293	27,000	0.18	2.0	3.3	2.38

**Table 2 materials-14-01440-t002:** Results comparison of present model with experimental prediction and other models.

Specimen Designation	The Threshold Pressure/MPa	Corrosion Rate/%	Time to Cover Cracking/yr				
			Experimental Value	Ref. [6]	Ref. [50]	Present Model	Error Value of Present Model to Experimental Results/%
S1	4.81	0.46	1.84	1.53–2.06	1.79	2.00	8.69
S2	5.17	0.48	3.54	3.34–4.49	3.10	3.51	0.85
S3	4.11	0.44	0.72	0.56–0.75	0.80	0.76	5.56
S4	5.06	0.55	2.38	1.79–2.40	2.25	2.16	9.24

**Table 3 materials-14-01440-t003:** The value of basic parameters. *d*_0_: thickness of porous zone; *i*_corr_: corrosion current densitiy.

Specimen Designation	2R/mm	*C*/mm	*d*_0_/m	*f*c/MPa	*E*c/MPa	*f*t/MPa	*i*_corr_/mA/cm^2^	Cracking Time/h	Date Source
A1	16	20	10	25	30,000	3.3	100	96	Andrade
B1	16	50	20	25	30,000	3.3	100	208	Alonso
B2	16	70	25	25	30,000	3.3	100	264	
C1	16	33	15	30	30,000	3.3	150	95	Maaddawy
D1	50	16	10	13.4	22,000	3.3	100	194.7	Vu
D2	25	16	10	28.8	30,000	3.3	100	116	

**Table 4 materials-14-01440-t004:** Results comparison of present model with experimental prediction and other models.

Specimen Designation	The threshold Pressure/MPa	Corrosion Rate/%	Time to Cover Cracking/h				
			Experimental Value	Ref. [16]	Ref. [34]	Present Model	Error Value of Present Model to Experimental Results/%
A1	3.68	0.35	96	84–119	82.08	106.55	10.99
B1	4.85	0.64	208	156–191	197.84	192.70	7.36
B2	5.17	0.79	264	203–237	281.35	239.18	9.40
C1	4.38	0.49	95	78–101	78.38	99.28	4.51
D1	1.95	0.21	194.7	-	172.68	194.37	1.70
D2	2.80	0.25	116	-	94.07	119.67	3.16

**Table 5 materials-14-01440-t005:** Results comparison of the present model with measuring prediction.

Specimen Designation	2R/mm	*C*/mm	*w/c* Ratio	*C_t_*/kg/m^3^	*f*c/MPa	Actual Time to Cracking/yr	Calculated Value/yr
Concrete column	10	20.35	0.40	11	20	0.26	0.33
Concrete walkway slab	12	25.0	0.40	11	20	0.44	0.42

**Table 6 materials-14-01440-t006:** The value of basic parameters to determine.

Basic Variables	Abbreviations	Value	Unit
Initial fracture toughness	*K* _Ic_ ^ini^	1.034	MPa·m^1/2^
Cracking fracture toughness	*K* _Ic_ ^un^	2.072	MPa·m^1/2^
The length of initial defects	*a*	2	mm
Chloride content	*C_t_*	4.8	kg/m^3^
Temperature	*T*	293	K
Relative humidity	*RH*	80	%
Concrete resistivity	*r*	10,000	Ohm·cm
Cover depth	*C*	30	mm
Diameter of steel bar	*D*	16	mm
Water-to-cement ratio	*w/c* ratio	0.45	-
Elastic Modulus	*E* _c_	2.7 × 10^4^	MPa
Tensile strength	*f* _t_	3.3	MPa
Poisson’s ratio	*vc*	0.18	-
Creep coefficient	*ϕ* _cr_	2.0	-
Expansion rates of corrosion products	*n*	2.0	-
Thickness of porous zone	*d* _0_	12.5	μm
Initial defect cracking size	*a*/*c*	0.20	-
Cover cracking size	*a*/*c*	0.72	-

**Table 7 materials-14-01440-t007:** The parameter values used in model verification of rust expansion cracking time.

Number	Basic Variables	Variable Name	Mean	Coefficient of Variation	Distribution Type	References
1	Cracking fracture toughness	*K* _Ic_ ^un^	2.072	0.177	Normal	[40]
2	Chloride content	*C_t_*	4.8 kg/m^3^	0.5	Normal	[52]
3	Temperature	*T*	293 K	0.2	Lognormal	[53]
4	Relative humidity	*RH*	75%	0.05	Normal	[56]
5	Cover depth	*C*	40 mm	0.2	Normal	[57]
6	Diameter of steel bar	*d*	16 mm	0.2	Lognormal	[58]
7	Water-to-cement ratio	*w/c* ratio	0.45	0.1	Lognormal	[52]
8	Elastic Modulus	*E* _c_	2.7 × 10^4^ MPa	0.12	Lognormal	[58]

## Data Availability

Data available in a publicly accessible repository that does not issue DOIs.

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
