# Peer review of "Modelling of Corrosion-Induced Concrete Cover Cracking Due to Chloride Attacking"

_materials, 2021, doi:10.3390/ma14061440_

Round 1

Reviewer 1 Report

Dear authors, thank you for your interesting paper. Corrosion of the reinforcement caused by the penetration of chloride ions into RC elements is a big problem everywhere in the world. Therefore, it is very important and useful to address this issue.

My comments are:

- line 10: reinforced concrete (RC) structures - please, first time use full nomenclature,

- introduction: there are many research works that deal with this issue (corrosion of reinforcement and crack in concrete) and could be mentioned and cited, for example:

  1. a) Brodňan, M., Koteš, P., VanÄ›rek, J., & Drochytka, R. (2017). Corrosion determination of reinforcement using the electrical resistance method. Materiali in Tehnologije, 51(1), 85-93. doi:10.17222/mit.2015.217

- line 106: missing denotation of angle beta?

- line 106: I think that here are missing the values of angle (Pi and 2Pi or in degrees),

- line 111: erase dot, or capital „A“ on the start of another sentence,

- line 121-122, formulas (3) and (4): I'm not sure if the expressions under the square root in equations (3) and (4) should be the same or different - it needs to be checked,

- line 123: where - not capital "W" – the same also in lines 130, 135, 138, 150, 191, 194, 224, 229, 236, 254, 268, 273, and 443,

- line 138: in text are missing KIcin and KIcun,

- line 162: in text are missing denotations of angle,

- line 172: The with capital "T" - start new sentence?

paragraph 2.1.2. – similar work focused on modelling of corrosion and its influence on cracks and resistance are also:

  • Koteš, P. (2013). Influence of corrosion on crack width and pattern in an RC beam. Paper presented at the Procedia Engineering, , 65 311-320. doi:10.1016/j.proeng.2013.09.048

- paragraph 2.1.2. – corrosion of reinforcement due to chlorides has mostly character of pit corrosion (local), but your presented/used model is surface corrosion, which is recommended to use for corrosion due to carbonization, why do you used model of surface corrosion?

- line 201, and 202 (formula (21)): corrosion rate is usually denotes as "r", but in formula (21) is denotation ro (Greek letter) - is it the same? Or is it anything else? Denotation “r” in text and “ro” in formulas of corrosion rate is also same later – line 218 and 219,

- line 223: is it "rcr" or "ro cr"? in the description below the equation is rcr,

- Line 224: unclear / incomprehensible - (g) is unit grams? so steel "quality" is steel weight? Steel quality of type of steel, denotation, or steel quality is given by yield strength,

- Line 224: rcr or rocr (ro - Greek letter) - in formula (29) is rocr, corrosion rate rcr has unit [micro meter/year], not [%],

- line 229: see above for equation (29),

- line 236: is it concrete chloride content on the surface of concrete (concrete cover)? or on surface of reinforcement? or in centre of gravity of the reinforcement?

- line 290, table 1: there is a column with denotation lower index “c” - only lower index "c"? what is it? Is it Poisson´s ration uc? Or another?

- line 290, table 1: last column - what is unit "a"? exposure period is time in unit [years], is not it?

- line 192, table 2: unit of time [a]? what is it? Years?

- line 303, table 3: column with denotation “Experimental value/h - what is it "experimental value" with unit [h]? Is it time in hours? What time? Of cracking? Or another?

- line 305: I do not understand, should it be "there is a good ...." or is it only "a good ..." without comma before?

- line 315, table 5, last two columns: "a" means hours, days, years?

- line 327-328: chloride content - where? on concrete surface?

- line 338, fig. 3c): please, specify, the corrosion rate has unit [micro m/year], this is in [%] - is it another type of corrosion rate?

- line 369, fig. 4: what is unit of the expansion rates of corrosion products? %?

- line 403: question as above - is it chloride content on concrete surface or chloride content on reinforcement surface (bottom point)?

- line 419: please, first time use full nomenclature, is it only TM method, or it is abbreviation,

- line 428: it is used to use the term "reliability margin"

- line 449: the reliability index and failure probability,

- line 452: lifetime,

- line 475: is it Monte Carlo method? please, first time use full nomenclature,

- line 480: is it Monte Carlo simulation? please, first time use full nomenclature,

- question: you investigated the time when occurred the crack on surface, but from the view of serviceability limit states, it is important to know the crack width and compare with limit value (occurring the crack on surface is not problem, crack width is a problem in structures. My question is – are you able to calculate the crack width?

Reviewer 2 Report

Dear Authors,

Congratulations on your research. Unfortunately several aspects need to be improved in your paper:

1. I strongly recommend an English grammar and style revision, there are several problems, such as: line 57 - initial defects initiation, probably you mean crack initiation, line 76 - the accuracy an rationality - perhaps accuracy and validity and so on.

2. some characters are missing in several equations (line 106, 161, 162) - perhaps an incompatibility given by the software, please check.

3.  Please keep a constant notation: as example, in line 201 you state that the corrosion rate is denoted "r" and in equation (21) you use the Greek letter "rho".

4. also, several parameters are repeated (a,b,c) throughout your work with different meanings and it can get confusing for the reader.

5. in eq. 33 and 34, given the current formatting, it could be miss understood that a matrix is involved

6. in the tables, the exposure period and actual time to cracking  is expressed as "a", please use an accepted SI unit.

My best regards.

Reviewer 3 Report

Dear authors, I attach a pdf with some comments. Please in case of resubmission provide your answers on this same file.

I find the article very interesting but very hard to read. Please make a thorough revision of the whole manuscript.

Round 2

Reviewer 3 Report

please find attached a pdf with some comments.

most important ones: i) you have units inconsistencies in some equations and ii) conclusions are too ambiguous in my opinion for such a long parametric article.

Round 3

Reviewer 3 Report

The authors have addressed all comments from the first and second review